# Survey of SNPs Associated with Total Number Born and Total Number Born Alive in Pig

**DOI:** 10.3390/genes11050491

**Published:** 2020-04-30

**Authors:** Siroj Bakoev, Lyubov Getmantseva, Faridun Bakoev, Maria Kolosova, Valeria Gabova, Anatoly Kolosov, Olga Kostyunina

**Affiliations:** 1Federal Science Center for Animal Husbandry named after Academy Member L.K. Ernst, Dubrovitsy 142132, Russian; siroj1@yandex.ru (S.B.); bakoevfaridun@yandex.ru (F.B.); m.leonovaa@mail.ru (M.K.); kolosov777@gmail.com (A.K.); kostolan@yandex.ru (O.K.); 2Department of Biology and Biotechnology, Southern Federal University, Rostov-on-Don 344006, Russia; gabova98@yandex.ru; 3Department of Biotechnology, Don State Agrarian University, Persianovski 346493, Russia

**Keywords:** pig, litter size, total number born alive, SNP (single nucleotide polymorphism), GWAS (Genome-Wide Association Studies)

## Abstract

Reproductive productivity depend on a complex set of characteristics. The number of piglets at birth (Total number born, Litter size, TNB) and the number of alive piglets at birth (Total number born alive, NBA) are the main indicators of the reproductive productivity of sows in pig breeding. Great hopes are pinned on GWAS (Genome-Wide Association Studies) to solve the problems associated with studying the genetic architecture of reproductive traits of pigs. This paper provides an overview of international studies on SNP (Single nucleotide polymorphism) associated with TNB and NBA in pigs presented in PigQTLdb as “Genome map association”. Currently on the base of Genome map association results 306 SNPs associated with TNB (218 SNPs) and NBA (88 SNPs) have been identified and presented in the Pig QTLdb database. The results are based on research of pigs such as Large White, Yorkshire, Landrace, Berkshire, Duroc and Erhualian. The presented review shows that most SNPs found in chromosome areas where candidate genes or QTLs (Quantitative trait locus) have been identified. Further research in the given direction will allow to obtain new data that will become an impulse for creating breakthrough breeding technologies and increase the production efficiency in pig farming.

## 1. Introduction

In the early 1990s the work on mapping the pig genome started and maps containing more than 1200 microsatellite markers appeared due to the development of the international project “PiGMap” and projects of the US Department of Agriculture and American agricultural institutes. These maps were used to identify the Quantitative Trait Loci (QTL) underlying the genetic architecture of pig productivity traits [1,2,3]. To date, an extended database has been created - Pig Quantitative Trait Locus Database (PigQTLdb), which presents 30170 QTLs for 688 pig trait of deferent classes [4].

The sows’ reproductive potential is the basis for continuous and efficient production. In recent decades, the BLUP (Best linear unbiased prediction) method has made a significant contribution to improving reproduction rates [5,6]. However, low heritability coefficients of reproductive traits and their sex-limited phenotypic exhibition lead to developing new approaches revealing the biological nature of reproductive performance. Great hopes are pinned on GWAS (Genome-Wide Association Studies) to solve the problems associated with studying the genetic architecture of reproductive traits of pigs [7,8,9,10].

GWAS results can be represented by the information on detected associations with various genetic aberrations: chromosomal mutations (whole chromosomes or their fragments), large inserts or deletions (100–100,000 nucleotides), small inserts or deletions (1 to 100), single nucleotide polymorphisms (SNP) [11]. Each SNP is represented by at least two alleles: minor (rarer) and major. Genetic variations with minor allele frequency exceeding 0.01% are numbered and assigned to rs index [12].

Currently SNP BeadChip technology is more affordable for genome-wide research than sequencing is. SNP BeadChips have been developed to include high (HD), medium (MD), or low (LD) genome distribution of markers. Rigid structure is noted as a disadvantage, which allows us to analyze only what is already predetermined by the BeadChip design and in this connection the potentially important information can be omitted [13]. Markers do not have the same density across all chromosomes and not fully track structural genetic variations such as insertions and deletions [13]. However, despite these shortcomings, SNP BeadChip has recently gained great popularity in studies of the genetic architecture of quantitative traits of farm animals and pigs in particular [14,15,16]. 

The first SNP BeadChips with a resolution of about 60 thousand markers covering all autosomal and X chromosome genes PorcineSNP60 BeadChip v2 were presented by the American company Illumina (San Diego, CA, USA). In addition to SNP BeadChip with high density, LD SNP BeadChip with low density have been proposed to reduce genotyping costs. Commercial LD SNP BeadChip «GeneSeek/Neogen GPP-Porcine LD Illumina Bead Chip panel» were also developed by «GeneSeek/Neogen» (Lincoln, NE, USA). Besides this, the company introduced HD SNP BeadChips with a higher density (about 70 thousand markers). HD SNP BeadChips containing about 650 thousand markers and including all markers of the «Illumina PorcineSNP60 BeadChip v2» array, were produced by «Affymetrix» (Santa Clara, CA, USA). It should be noted that SNP BeadChip can be custom made including specific SNPs associated with given traits of productivity.

Reproductive productivity depend on a complex set of characteristics. The number of piglets at birth (Total number born, Litter size, TNB) and the number of alive piglets at birth (Total number born alive, NBA) are the main indicators of the reproductive productivity of sows in pig breeding [17,18,19,20,21]. These indicators reflect the level of all physiological processes associated with fertilization, intrauterine development of the fetus and a sow’s labor, and are also quite easy to account for.

## 2. SNPs Associated with Total Number Born Alive

Currently on the base of Genome map association results 88 SNPs associated with NBA have been identified and presented in the Pig QTLdb database (Table 1). The results are based on a study of pigs such as Large White, Large White, Yorkshire, Landrace [7,14,22,23,24,25,26] Duroc [27] and Erhualian [28]. SNPs, associated with the NBA are represented in all Sus scrofa chromosomes (SSC) except SSCY. 

Search for SNPs associated with NBA of Chinese Erhualian pigs was carried out by Ma et al. [28]. Sows with high and low estimated breeding values (EBVs) were selected for genotyping. According to the research results, 9 SNPs associated with Pig QTLdb with NBA were presented. The greatest effect was found for SNP rs81447100 (SSC13), which was additionally tested on Erhualian pigs (*n* = 313), Sutai (*n* = 173) and Yorkshire (*n* = 488). In all groups under study, a significant association between allelic variants of SNP rs81447100 (SSC13) and NBA was determined. However, allele A was desirable for the Erhualian pigs and allele G for the Sutai and Yorkshire pigs. 

According to research work of Wu et al. [14], conducted on Landrace and Yorkshire pigs the database contains 15 SNPs localized on SSC1 and SSC8 and associated with NBA. 11 of these SNPs are located in the QTL regions annotated earlier, and 4 SNPs are presented for the first time. All these 4 SNPs (rs329624627, rs339929690, rs322202112 and rs330585697) are located on the SSC1. The most significant effect was established for SNP rs332924521 (SSC1).

Coster et al. [23] conducted associative studies on Large White pigs from two commercial lines of «Hypor» and «Topigs», and revealed 4 SNPs associated with NBA located on SSC7 (rs81397142 and rs81397215), SSC1 (rs81348779) and SSC2 (rs81356698).

Bergfelder-Drüing et al. [7] conducted research using Large White and Landrace pigs. Preliminary calculations based on graphs of multidimensional scaling showed the genetic distance between the breeds. For analysis pigs were divided into two clusters taking into account the breed and intra-breed clusters taking into account the breeding economy. As a result, 17 SNPs associated with NBAs were identified, 5 of these SNPs had a minor allele frequency less than 1%. The study of Bergfelder-Drüing et al. [7] was the first to show an association with NBA for SNP rs81430147 (SSC11). All other SNPs were found in chromosome regions where candidate genes or QTLs affecting pig reproductive traits have already been identified. It should be noted that different SNPs were established for each breed cluster, and no associative communications were established simultaneously in two breeds. 13 SNPs were identified for Large White sows, and 4 SNPs for Landrace sows. In sows Landrace SNPs are localized on SSC7, SSC9, SSC11 and SSC16. Large White on SSC3, SSC5, SSC9, SSC10, SSC11 and SS18. The most significant effects on NBA of Large White sows are found for 4 SNPs: rs81379421 (SSC3), rs81417393 (SSC9), rs81242222 (SSC11) and rs81469701 (SSC18).

Research carried out by Wang et al. [22] on Large White pigs identified 6 SNPs associated with NBA located on SSC2, SSC3, SSC13, SSC14 and SSC18. The most significant effect was determined for SNP rs334867206 (SSC3), according to which pigs of AA genotype had more NBA, compared with analogues of GG genotype. 

Suwannasing et al. [26] investigated Large White and Landrace pigs and established 25 SNPs for the NBA. Of these, 11 SNPs located in SSC1 and SSCX were defined for Large White pigs, 14 SNPs for Landrace pigs in SSC2 and SSC6. It is remarkable that SNPs on SSCX showed significance only at the NBA for Large White sows.

In the course of research on commercial pigs, Li et al. [25] established 2 SNPs (rs342908929 (SSC6) and rs324003968 (SSC15)) associated with NBA. According to the results of Chen et al. [27] obtained from the sows of Duroc in the database presents 9 SNPs and all of them are localized on SSC6. 

An et al. [24] studied the *IGFBP2* (SSC15) and *IGFBP3* (SSC18) genes in Berkshire pigs. Their results showed significant SNPs in these genes. This work also analyzed the expression levels of *IGFBP2* and *IGFBP3* mRNA in the endometrium in pigs of various genotypes. Homozygous GG pigs expressed higher levels of *IGFBP3* mRNA in the endometrium than pigs of other genotypes, and a positive correlation was observed between litter size traits and *IGFBP3* but not *IGFBP2* expression level. These results suggest that SNPs in the *IGFBP2* and the *IGFBP3* gene are useful biomarkers for the little traits of pigs. According to the results of this work 2 SNPs are included in PigQTLdb rs45435330 on SSC15 as Genome map association for NBA and TNB.

## 3. SNPs Associated with Total Number Born

In general, in Pig QTLdb for TNB showed 218 SNPs (as Genome map association), of which 155 SNPs were detected by He et al. [8] and Ma et al. [28] in Erhualians, 52 SNPs are defined in Large White, Yorkshire, Landrace, 1 SNP in Berkshire and 10 SNPs in Duroc (Table 2) [14,18,22,23,24,25,29,30,31,32,33]. SNPs associated with TNB are represented in all Sus Scrofa Chromosome except for SSCY. 

For an associative study He et al. [8] selected Erhualan sows with high and low EBV values. According to the results of their work, the most significant SNPs were detected on SSC2 chromosomes (rs81367039), SSC7 (rs80891106), SSC8 (rs81399474), SSC12 (rs81434499), SSC14 (rs80938898). Among them SNPs on chromosomes SSC2, SSC7, and SSC12 were annotated for the first time. To study the effect of significant SNPs additional studies were conducted on a livestock of 331 Erhualan sows. According to the results of additional testing, significant differences in TNB were found only for SNP rs81399474 (SSC8). In the studies of Ma et al. [28] 8 SNPs were identified as associated with TNB in Erhualian sows and located on SSC1, SSC4, SSC7, SSC8, SSC10, SSC12, SSC13and SSC16.

Studies of Large White pigs conducted by Sell-Kubiak et al. [30] allowed the identification of 10 SNPs associated with the number of piglets at birth and located on SSC1, SSC5, SSC8, SSC11, SSC13 and SSC18. SNPs (rs80989787 and rs81289355) located on SSC11 were annotated for the first time in this paper. The most significant effect has been determined for SNP rs80989787 (SSC11). According to the results of associative studies conducted by Uimari et al. [18] on Finnish Landrace pigs 10 SNPs were identified. All established SNPs are located on SSC9.

The most significant effect was established for SNP rs81300575 (SSC9), which amounted to about 1 piglet between two homozygous genotypes. In the studies of Uimari et al. [18] it was also noted that in the past 15 years the frequency of the desired SNP rs81300575 (SSC9) genotype in the studied population has increased from 0.14 to 0.22.

Zhang et al. [29] conducted research on Duroc pigs and identified 10 SNPs associated with TNB. The most significant SNPs were rs80979042 and rs80825112 located on SSC14. In addition, the remaining potential SNPs were located on SSC5, SSC6, SSC12 and SSC17.

In the studies of Coster et al. [23], Wu et al. [14], Wang et al. [22] and Li et al. [25] the effects of SNPs on TNBs were investigated along with the search of SNPs associated with NBAs. So according to the results of Coster et al. [23] the database contains 16 SNPs defined on SSC1, SSC2, SSC7, SSC14 and SSC18. Wu et al. [14] established 5 SNPs associated with TNB located on SSC8 and SSC14. Wang et al. [22] identified 11 SNPs for TNB on SSC1-SSC5, SSC13 and SSC18, and Li et al. [25] 2 SNPs (rs342908929 (SSC6) and rs324003968 (SSC15)). Besides this, Pig QTLdb presents SNPs associated with TNB according to the studies of Onteru et al. [31]—SNP rs81452018 SSC15, Wang et al. [32]—SNP rs345476947 SSC6 and Liu et al. [33]—rs rs55618224 SSC6. 

## 4. Potential Candidate Genes for Litter Traits of Pig

Accordingly, Pig QTLdb presents 306 SNPs associated with TNB (218 SNPs) and NBA (88 SNPs) in pigs of various breeds. We should note that 12 SNPs out of 306 SNPs provided in PigQTLdb as a Genome map association with TNB and NBA, are presented twice (add Appendix A). Perhaps this is due to the fact that the connection of SNPs with TNB and NBA was established in one study, for example, as for SNPs rs81348779 on SSC1 and rs81356698 on SSC2 [23]. These SNPs are localized in the intron of the *UBE3A* (SSC1) and *EIF3M* (SSC2) genes. The *UBE3A* gene encodes *ubiquitin protein ligase E3A*, which plays an essential role in the normal development and functioning of the nervous system, and helps regulate the balance between proteostasis synthesis and degradation in the joints between synapses. Human and mouse *UBE3A* is maternal imprinted [34]. However, there is no precise information regarding imprinting of pig *UBE3A*. According to the study performed by Wang et al. [35] *UBE3A* was not imprinted in the skeletal muscle of neonate pigs of Landrace boars and Laiwu sows cross. Further research will probably provide more information to clarify the effects of the *UBE3A* gene and its and its relation to pig fecundity. The *EIF3M* gene (the eukaryotic translation initiation factor 3 subunit M) is a complex translation initiation factor consisting of 13 subunits (*EIF3A-EIF3M*) and which is involved in mRNA modulation [36]. The *EIF3* complex is necessary for the key stages of protein synthesis initiation [37]. Previous studies have shown that *EIF3M* encodes a protein that is critical for mouse embryonic development [38]. 

In studies performed by Wang et al. [22] a relationship with TNB and NBA was defined for SNPs rs334867206 (intergenic_variant) and rs319494663 (upstream gene variant *ssc-let-7a-2*) on SSC3 and rs81471172 (intron variant *HECW1*) on SSC18. *Sus scrofa let-7a-2 stem-loop (ssc-let-7a-2*) belongs to miRNAs, a class of small non-coding RNAs (~21 nt) that regulate the mRNAs translation on the post-transcriptional level, mainly by binding their targets with the three prime untranslated region (3’-UTR) [39]. A variety of studies have shown that miRNAs can play a potential regulatory role in porcine ovary, testis and spermatogenesis [40,41,42].

The *HECW1* gene, also known as *NEDD4-like ubiquitin protein ligase 1* (*NEDL1*), is expressed in human neuronal tissues and enhances p53-mediated apoptotic cell death [43]. Supposedly, it regulates the bone morphogenetic protein signaling pathway during embryonic development and bone remodeling [44]. In the work of Li et al. [25] associations with TNB and NBA are defined for SNP rs342908929, which is localized on SSC6 in the intron of the *ZFYVE9* gene (*zinc finger FYVE-type containing 9 domain*). The protein encoded by *ZFYVE9* is involved in the signaling pathway of *transforming growth factor-beta* (*TGFB*) and directly interacts with *SMAD2* and *SMAD3*, needed for normal follicular development and ovulation [45]. 

In the studies performed by Ma et al. [28] a relation with TNB and NBA was defined for SNPs rs80882306 (intergenic variant) on SSC7 and rs319494663 (intron variant PARD3) on SSC10. *PARD3* (*PAR-3*) is a *scaffold-like PDZ-containing protein*. *PAR-3* forms a complex with *PAR-6* and *atypical protein kinase C* (*PAR-3-atypical protein kinase C-PAR-6 complex*) and is associated with the establishment of cell polarization [46,47,48]. McCole [49] argued that mutations in *PARD3* can also influence the recovery of wounds by weakening the response of the epithelial barrier to a damage or inflammation. Concerning its significant role in the regulation of various stages of ovarian development and the control of steroidogenesis in a ripening follicle An et al. [24] studied the *IGFBP-2* gene (SSC15) and defined SNP rs45435330 associated with TNB and NBA [50]. 

But more interesting are the variants repeated in two independent studies. Thus, rs80927364 (intron variant *DDAH1*) on SSC4, rs81434499 (intergenic variant) on SSC12, rs81447100 (intron variant *CLSTN2*) on SSC13 showed significant associations with TNB and NBA in the studies of He et al. [8] and Ma et al. [28] respectively. 

The *DDAH1* gene (dimethylarginine dimethylaminohydrolase 1), along with other *DDAH* genes, is involved in the metabolic control of asymmetric dimethylarginine (ADMA), contributes to the maintenance of vascular homeostasis due to the expansion of blood vessels, suppression of inflammation and inhibiting vascular smooth muscle cells, adhesion and aggregation of platelets [51,52,53]. Data on the metabolic control of ADMA by *DDAH* genes and their effect on endothelial cells were obtained in animal studies. Transgenic mice with overexpression of *DDAH1* showed a twofold decrease of ADMA in plasma, associated with a twofold increase of NOS activity in tissue [54]. Conversely, the *DDAH1* Knock Out Mice exhibited increased pulmonary endothelial permeability as a result of increased ADMA, which was prevented by over-expression of *DDAH1* and *DDAH2* in endothelial cells [55].

*CLSTN2* (calsintenin 2) is associated with obesity in mammals, especially in the process of increasing adipocytes in visceral tissues and in subcutaneous fat [56]. Santana et al. [57] defined *CLSTN2* as a candidate gene associated with ultrasound-derived measurements of the rib-eye area, backfat thickness and rumpfat thickness in Nellore cattle. Adipose tissue is an active endocrine organ and proteins involved in forming adipose tissue is increasingly attracting attention as mammalian reproduction markers [58,59].

We can also note SNPs rs81463092 (intergenic variant) and rs81289648 (intergenic variant) defined in the works of He et al. [8] and Ma et al. [28], localized on SSC16 and lying in close proximity to each other (68 kb). The table shows the intervals between neighboring SNPs (add Appendix A). Hence, we can note a number of closely related SNPs identified in various studies. For example, rs55618224 (3 prime UTR variant) and rs81388947 (intergenic variant) on SSC6 are defined by Liu et al. [33] and Chen et al. [27] and the interval between them is 140 kb. SNP rs336670754 (synonymous variant *ARID1A*) and SNP rs329711941 (intron variant *ZDHHC18*) are located at a distance of about 39 kb from each other, defined on SSC6 by Chen et al., (2019) and Zhang et al. [29] respectively. 

On the whole, we can note that SNPs presented in PigQTLdb as associations with TNB and NBA are more localized in genes (intron variant 60%). Wherein that, 15% are intergenic variants and this can be considered as an evidence of significance for intergenic variants in the genetic architecture of reproductive traits (add Appendix A).

In conclusion, we wanted to estimate the gene-based protein-protein interactions obtained with the studied SNPs being localized. The total number of genes was 127, but only 40 genes formed some pairs or chains (Figure 1). It is interesting that some chains are composed of genes (SNPs) identified in only one study (add Appendix A). For example, *RBP7*, *LRRK1*, *UBE4B, TRPC5*, *LHFPL1* genes were identified by Suwannasing et al. [26]; *DLC1*, *SEMA3A*, *DPYSL3*, *NRPI* by He et al. [8] similarly. However, the other chains includes genes identified in various studies such as *GRIA4* [8], *CLSTN2* [8], *GRIP1* [28], *GABRA5* [22], *UBE3A* [22], *COPG2* [22], *ANKRD40* [8], *MYO10* [7], *SHISA9* [8].

The presented review shows that most SNPs annotated from genome-wide studies are found in chromosome regions where candidate genes or QTLs have already been identified. Future research will be aimed at annotating sequences and analyzing these data, which can contribute to better understanding the mechanisms of reproductive traits formation.

However, the main problem in summarizing the results is the design of the experiment (consideration of traits, features of the studied populations, etc.). Some of the factors affecting the result are not controlled by the researcher (such as the trait inheritance level, the genetic architecture of the population, linkage disequilibrium, etc.). On the other hand, control over the reference data design (the choice of markers, the model for evaluating effects, etc.) that affects the accuracy of the results is controlled by the researcher. In the analyzed studies the majority of genotyping works were done by using Illumina PorcineSNP60 BeadChip or GeneSeek PorcineSNP80 BeadChip. There are numerous statistical approaches to conducting GWAS. However, the mixed model is more preferable, being implemented in various software packages (GEMMA, ASREML, GenABEL, etc.). There is still no consensus on the best method. Many researchers emphasize that the defined associations and their significance depend on the methodologies and details of data analysis, and we need to develop statistical approaches in order to improve the accuracy of the obtained information [16,19].

## 5. Conclusions

The results of genome studies show the prospects of this approach to studying the genetic architecture of reproductive indicators in pigs. The material presented in this overview can be used in developing local test systems for a limited number of SNPs to estimate their effect on the own livestock. In addition, gene networks can be built on the basis of the presented SNPs to find potential candidate genes for reproductive signs. On the whole, these results help us to understand the genetic basis of pig reproductive traits and can be used in further studies. Further research in this direction will provide new data that will be a powerful impetus for creating breakthrough breeding technologies and improving the efficiency of breeding production in pig farming.

## Figures and Tables

**Figure 1 genes-11-00491-f001:**
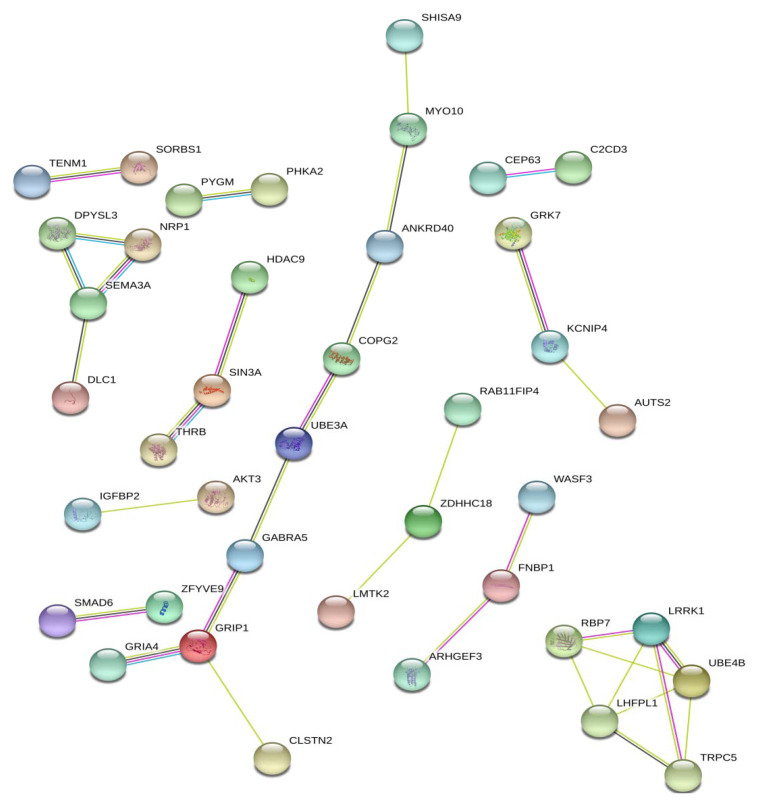
Protein-protein interactions obtained with the studied SNPs (STRING db).

**Table 1 genes-11-00491-t001:** SNPs (Single nucleotide polymorphism) associated with NBA (Total number born alive) presented in the PigQTLdb (Pig Quantitative Trait Locus Database). Ensembl db, Sus Scrofa 11.1 (NA—This variant has not been mapped on Ensembl db, Sus Scrofa 11.1).

	SNP	SSC	Location	Allele	Consequence	SYMBOL	Trait	Breed	Reference
1	rs339929690	1	59515751	C	intergenic_variant	-	NBA	Landrace/Yorkshire	[14]
2	rs329624627	1	72605122	C	intron_variant	CRYBG1	NBA	Landrace/Yorkshire	[14]
3	rs338462676	1	74750265	T	intron_variant	-	NBA	Landrace/Yorkshire	[14]
4	rs80982567	1	99668135	G	intron_variant	-	NBA	Landrace/Yorkshire	[14]
5	rs329761313	1	116715803	C	intron_variant	-	NBA	Landrace/Yorkshire	[14]
6	rs329145797	1	116791808	T	intron_variant	-	NBA	Landrace/Yorkshire	[14]
7	rs80830052	1	139481542	A	intron_variant	ALDH1A3	NBA	Landace/Large White	[26]
8	rs80930659	1	139579572	G	intron_variant	LRRK1	NBA	Landace/Large White	[26]
9	rs80804265	1	139608452	A	intron_variant	LRRK1	NBA	Landace/Large White	[26]
10	rs80862569	1	139636710	C	intergenic_variant	-	NBA	Landace/Large White	[26]
11	rs80846651	1	139655026	A	intergenic_variant	-	NBA	Landace/Large White	[26]
12	rs81348779	1	141989297	G	intron_variant	UBE3A	NBA	Large White	[22]
13	rs329931325	1	144835089	T	intergenic_variant	-	NBA	Landrace/Yorkshire	[14]
14	rs326961952	1	145395812	G	intergenic_variant	-	NBA	Landrace/Yorkshire	[14]
15	rs336474421	1	145452726	T	intergenic_variant	-	NBA	Landrace/Yorkshire	[14]
16	rs322202112	1	146334522	C	intron_variant	-	NBA	Landrace/Yorkshire	[14]
17	rs332924521	1	239558816	C	intron_variant	XPA	NBA	Landrace/Yorkshire	[14]
18	rs330585697	1	239775818	T	intron_variant	TRMO	NBA	Landrace/Yorkshire	[14]
19	rs334029855	1	NA				NBA	Landrace/Yorkshire	[14]
20	rs81291755	2	20563683	G	intergenic_variant	-	NBA	Landrace/Yorkshire	[26]
21	rs81355894	2	20637563	C	intergenic_variant	-	NBA	Landace/Large White	[26]
22	rs81355903	2	20665892	T	intergenic_variant	-	NBA	Landace/Large White	[26]
23	rs81355915	2	20717076	G	intergenic_variant	-	NBA	Landace/Large White	[26]
24	rs81356698	2	28380318	G	intron_variant	EIF3M	NBA	Large White	[22]
25	rs81265647	2	125660328	T	upstream_gene_var	-	NBA	Landace/Large White	[26]
26	rs328177895	2	NA				NBA	Large White	[23]
27	rs81379421	3	27307613	G	intron_variant	XYLT1	NBA	Landace/Large White	[7]
28	rs334867206	3	43312168	T	intergenic_variant	-	NBA	Large White	[23]
29	rs319494663	3	43463318	C	upstream_gene_var	ssc-let-7a-2	NBA	Large White	[23]
30	rs80927364	4	130597207	A	intron_variant	DDAH1	NBA	Erhualian	[28]
31	rs80890206	5	31022891	C	upstream_gene_var	GRIP1	NBA	Erhualian	[28]
32	rs80867243	5	87337099	C	intron_variant	ELK3	NBA	Landace/Large White	[7]
33	rs81279319	5	NA				NBA	Landace/Large White	[7]
34	rs81383147	5	NA				NBA	Landace/Large White	[7]
35	rs81320475	6	70313133	A	upstream_gene_var	RBP7	NBA	Landace/Large White	[26]
36	rs81285644	6	70323076	G	intron_variant	RBP7	NBA	Landace/Large White	[26]
37	rs81275494	6	70408106	G	intron_variant	UBE4B	NBA	Landace/Large White	[26]
38	rs81279050	6	70418172	A	intron_variant	UBE4B	NBA	Landace/Large White	[26]
39	rs81270030	6	70428427	T	intron_variant	UBE4B	NBA	Landace/Large White	[26]
40	rs81388947	6	80229040	A	intergenic_variant	-	NBA	Duroc	[27]
41	rs704072370	6	80388745	A	intergenic_variant	-	NBA	Duroc	[27]
42	rs81332505	6	83142163	A	intron_variant	MAN1C1	NBA	Duroc	[27]
43	rs81476037	6	83545020	T	intron_variant	PDIK1L	NBA	Duroc	[27]
44	rs328276462	6	83863384	G	downstream_gene_var	HMGN2	NBA	Duroc	[27]
45	rs335265547	6	83871998	C	upstream_gene_var	-	NBA	Duroc	[27]
46	rs81287462	6	83986459	A	intergenic_variant	-	NBA	Duroc	[27]
47	rs81332455	6	84069079	G	intron_variant	ARID1A	NBA	Duroc	[27]
48	rs336670754	6	84118086	C	synonymous_variant	ARID1A	NBA	Duroc	[27]
49	rs342908929	6	159933806	T	intron_variant	ZFYVE9	NBA	Commercia	[25]
50	rs81345088	6	168897980	A	intron_variant	ZMYND12	NBA	Landace/Large White	[26]
51	rs81259198	6	168899114	A	intron_variant	ZMYND12	NBA	Landace/Large White	[26]
52	rs81245903	6	168916590	T	intron_variant	RIMKLA	NBA	Landace/Large White	[26]
53	rs81273774	6	168931165	C	intron_variant	RIMKLA	NBA	Landace/Large White	[26]
54	rs80882306	7	11003222	G	upstream_gene_var	-	NBA	Erhualian	[28]
55	rs325729252	7	109101108	C	intergenic_variant	-	NBA	Landace/Large White	[7]
56	rs81397142	7	NA				NBA	Large White	[22]
57	rs81397215	7	NA				NBA	Large White	[22]
58	rs318557169	8	95381096	G	intergenic_variant	-	NBA	Landrace/Yorkshire	[14]
59	rs327655683	8	95402330	A	intergenic_variant	-	NBA	Landrace/Yorkshire	[14]
60	rs81257618	9	13492371	A	intergenic_variant	-	NBA	Landace/Large White	[7]
61	rs81417393	9	126541658	C	intron_variant	-	NBA	Landace/Large White	[7]
62	rs81421148	10	16506621	C	intron_variant	AKT3	NBA	Landace/Large White	[7]
63	rs80978601	10	25256004	G	intergenic_variant	-	NBA	Landace/Large White	[7]
64	rs81255997	10	28330355	C	intergenic_variant	-	NBA	Landace/Large White	[7]
65	rs81274366	10	57593745	G	intron_variant	PARD3	NBA	Erhualian	[28]
66	rs81236069	10	58288430	A	intergenic_variant	-	NBA	Landace/Large White	[7]
67	rs81302230	10	58290108	G	intergenic_variant	-	NBA	Landace/Large White	[7]
68	rs81430147	11	836702	C	downstream_gene_var	-	NBA	Landace/Large White	[7]
69	rs81430859	11	4296303	C	intron_variant	WASF3	NBA	Landace/Large White	[7]
70	rs81242222	11	67129570	A	intergenic_variant	-	NBA	Landace/Large White	[7]
71	rs81434499	12	36207006	G	intergenic_variant	-	NBA	Erhualian	[28]
72	rs80962240	13	52784022	C	intron_variant	FOXP1	NBA	Large White	[23]
73	rs81215583	13	71905692	C	synonymous_variant	RPN1	NBA	Erhualian	[28]
74	rs81447100	13	80866929	A	intron_variant	CLSTN2	NBA	Erhualian	[28]
75	rs81447231	13	82603578	A	intron_variant	GRK7	NBA	Erhualian	[28]
76	rs319258722	14	123766842	T	intergenic_variant	-	NBA	Large White	[23]
77	rs45435330	15	118847390	A	intron_variant	IGFBP2	NBA	Berkshire	[24]
78	rs324003968	15	NA				NBA	Commercia	[25]
79	rs81459590	16	6025546	A	intron_variant	MYO10	NBA	Landace/Large White	[7]
80	rs80952566	17	27123793	T	intron_variant	-	NBA	Erhualian	[28]
81	rs81469701	18	42920811	T	upstream_gene_var	SCRN1	NBA	Landace/Large White	[7]
82	rs81471172	18	51640938	T	intron_variant	HECW1	NBA	Large White	[23]
83	rs81473442	X	91880535	C	intron_variant	TRPC5	NBA	Landace/Large White	[26]
84	rs81323503	X	92070342	A	intergenic_variant	-	NBA	Landace/Large White	[26]
85	rs81283192	X	92244402	A	intron_variant	RTL4	NBA	Landace/Large White	[26]
86	rs337547716	X	92330719	A	intron_variant	RTL4	NBA	Landace/Large White	[26]
87	rs80834138	X	92447181	A	intron_variant	LHFPL1	NBA	Landace/Large White	[26]
88	rs81339510	X	118854990	T	intergenic_variant	-	NBA	Landace/Large White	[26]

**Table 2 genes-11-00491-t002:** SNPs (Single nucleotide polymorphism) associated with TNB (Total number born alive) presented in the Pig QTLdb (Pig Quantitative Trait Locus Database). Ensembl db, Sus Scrofa 11.1 (NA—This variant has not been mapped on Ensembl db, Sus Scrofa 11.1).

NO	SNP	SSC	Location	Allele	Consequence	SYMBOL	Trait	Breed	Reference
1	rs80972494	1	7262020	C	intergenic_variant	-	TNB	Large White	[22]
2	rs80787893	1	14869534	C	intron_variant	ZBTB2	TNB	Large White	[30]
3	rs80851003	1	20072414	G	intergenic_variant	-	TNB	Erhualian	[8]
4	rs80970692	1	97934013	A	intron_variant	ZBTB7C	TNB	Large White	[23]
5	rs80845110	1	139993309	A	intron_variant	GABRG3	TNB	Large White	[22]
6	rs80933698	1	140097794	G	intron_variant	GABRG3	TNB	Large White	[22]
7	rs80869858	1	140178409	T	intron_variant	GABRG3	TNB	Large White	[22]
8	rs80999701	1	140290677	A	intron_variant	GABRG3	TNB	Large White	[22]
9	rs81348717	1	140463533	G	intron_variant	GABRG3	TNB	Large White	[22]
10	rs81348724	1	140550299	A	intron_variant	GABRA5	TNB	Large White	[22]
11	rs81348751	1	140808020	A	intron_variant	GABRB3	TNB	Large White	[22]
12	rs80989931	1	140896691	A	intron_variant	GABRB3	TNB	Large White	[22]
13	rs81348779	1	141989297	G	downstream_gene_var	UBE3A	TNB	Large White	[22]
14	rs80912860	1	164637575	C	intergenic_variant	-	TNB	Large White	[30]
15	rs80956812	1	164674664	A	intron_variant	SMAD6	TNB	Large White	[30]
16	rs81267574	1	222278217	G	intron_variant	PIP5K1B	TNB	Erhualian	[8]
17	rs80938435	1	245222778	T	intergenic_variant	-	TNB	Erhualian	[8]
18	rs81296573	1	263022859	A	intergenic_variant	-	TNB	Erhualian	[8]
19	rs80913204	1	270166629	T	intron_variant	FNBP1	TNB	Erhualian	[8]
20	rs81367039	1	NA				TNB	Erhualian	[8]
21	rs81330112	2	3058633	A	upstream_gene_var	CTTN	TNB	Erhualian	[8]
22	rs81214065	2	7420349	A	missense_variant	PYGM	TNB	Erhualian	[8]
23	rs81474834	2	9252507	T	intron_variant	-	TNB	Erhualian	[8]
24	rs81273273	2	24914867	G	intron_variant	LDLRAD3	TNB	Erhualian	[8]
25	rs81356698	2	28380318	G	downstream_gene_var		TNB	Large White	[22]
26	rs346316162	2	84269464	T	intron_variant	ANKRD31	TNB	Erhualian	[8]
27	rs81360234	2	85003252	G	intron_variant	SV2C	TNB	Erhualian	[8]
28	rs81270902	2	136051929	G	intron_variant	FSTL4	TNB	Erhualian	[8]
29	rs81307772	2	142675693	C	intron_variant	PCDHAC2	TNB	Erhualian	[8]
30	rs81366836	2	145019988	C	intergenic_variant	-	TNB	Erhualian	[8]
31	rs81367208	2	146880995	T	intergenic_variant	-	TNB	Erhualian	[8]
32	rs81346993	2	148668538	G	intron_variant	DPYSL3	TNB	Erhualian	[8]
33	rs328177895	2	NA				TNB	Large White	[23]
34	rs81367039	2	NA				TNB	Erhualian	[8]
35	rs81367039	2	NA				TNB	Erhualian	[8]
36	rs334519198	3	5376605	T	intron_variant	LMTK2	TNB	Large White	[23]
37	rs81243084	3	14585248	T	intron_variant	AUTS2	TNB	Erhualian	[8]
38	rs81318451	3	20859502	T	intron_variant	HS3ST4	TNB	Erhualian	[8]
39	rs81319541	3	23688143	G	downstream_gene_var	IGSF6	TNB	Erhualian	[8]
40	rs81379942	3	30034842	A	intron_variant	SHISA9	TNB	Erhualian	[8]
41	rs81369361	3	43227069	A	intergenic_variant	-	TNB	Large White	[23]
42	rs334867206	3	43312168	T	intergenic_variant	-	TNB	Large White	[23]
43	rs319494663	3	43463318	C	upstream_gene_variant	ssc-let-7a-2	TNB	Large White	[23]
44	rs338135576	3	52266739	T	intron_variant	IL1R1	TNB	Erhualian	[8]
45	rs81370592	3	53699303	C	intergenic_variant	-	TNB	Erhualian	[8]
46	rs81377897	3	124275280	G	upstream_gene_variant	-	TNB	Erhualian	[8]
47	rs81272059	3	125892592	G	intron_variant	NOL10	TNB	Erhualian	[8]
48	rs81319839	4	18194352	A	intergenic_variant	-	TNB	Large White	[23]
49	rs81312912	4	18196598	A	intergenic_variant	-	TNB	Large White	[23]
50	rs80986621	4	87990037	C	missense_variant	UAP1	TNB	Erhualian	[8]
51	rs80860510	4	89154886	A	intron_variant	SDHC	TNB	Erhualian	[8]
52	rs80987610	4	89242132	G	downstream_gene_var	APOA2	TNB	Erhualian	[8]
53	rs80910021	4	96557774	C	intergenic_variant	-	TNB	Erhualian	[8]
54	rs80999559	4	98260992	C	downstream_gene_var	CERS2	TNB	Erhualian	[8]
55	rs80927364	4	130597207	A	intron_variant	DDAH1	TNB	Erhualian	[8]
56	rs81385465	5	4131296	G	intergenic_variant	-	TNB	Large White	[30]
57	rs336638152	5	11274122	A	intron_variant	-	TNB	Duroc	[29]
58	rs327336155	5	31752997	G	intergenic_variant	-	TNB	Large White	[23]
59	rs80890539	5	49084918	T	intergenic_variant	-	TNB	Erhualian	[8]
60	rs80999110	5	65577418	T	intergenic_variant	-	TNB	Duroc	[29]
61	rs328217833	5	78953555	C	intergenic_variant	-	TNB	Erhualian	[8]
62	rs81303269	5	79020148	A	intergenic_variant	-	TNB	Erhualian	[8]
63	rs81236331	6	4395948	T	intron_variant	WFDC1	TNB	Erhualian	[8]
64	rs81307446	6	4817058	T	intron_variant	CDH13	TNB	Erhualian	[8]
65	rs81393472	6	21728055	A	intergenic_variant	-	TNB	Erhualian	[8]
66	rs345476947	6	54042595	T	intron_variant	FUT2	TNB	Large White	[32]
67	rs81322640	6	74250906	T	intron_variant	KAZN	TNB	Erhualian	[8]
68	rs81283746	6	78467732	C	downstream_gene_var	UBXN10	TNB	Erhualian	[8]
69	rs55618224	6	80088638	C	3_prime_UTR_variant	-	TNB	Yorkshire	[33]
70	rs81318862	6	82375634	A	intergenic_variant	-	TNB	Duroc	[29]
71	rs329711941	6	84156205	C	intron_variant	ZDHHC18	TNB	Duroc	[29]
72	rs81391439	6	129931172	A	intergenic_variant	-	TNB	Erhualian	[8]
73	rs342908929	6	159933806	T	intron_variant	ZFYVE9	TNB	Commercia	[25]
74	rs81319462	6	163598597	G	intergenic_variant	-	TNB	Erhualian	[8]
75	rs81319428	6	163598619	C	intergenic_variant	-	TNB	Erhualian	[8]
76	rs80882306	7	11003222	G	upstream_gene_variant	-	TNB	Erhualian	[28]
77	rs80813007	7	21583296	G	downstream_gene_var	-	TNB	Erhualian	[8]
78	rs80938431	7	22008244	G	intron_variant	-	TNB	Erhualian	[8]
79	rs80933422	7	27905935	T	intergenic_variant	-	TNB	Erhualian	[8]
80	rs80797074	7	29268803	G	synonymous_variant	DST	TNB	Erhualian	[8]
81	rs340672537	7	38439290	T	intron_variant	ABCC10	TNB	Erhualian	[8]
82	rs81398070	7	58232894	C	intron_variant	SIN3A	TNB	Erhualian	[8]
83	rs80824208	7	63161681	G	intron_variant	SLC25A21	TNB	Erhualian	[8]
84	rs326644823	7	63618208	A	downstream_gene_var	-	TNB	Erhualian	[8]
85	rs80867596	7	70133128	G	downstream_gene_var	-	TNB	Erhualian	[8]
86	rs80891106	7	73467314	A	intergenic_variant	-	TNB	Erhualian	[8]
87	rs81295302	7	74333495	T	intron_variant	STXBP6	TNB	Erhualian	[8]
88	rs81398127	7	75705436	G	3_prime_UTR_variant	CMTM5	TNB	Erhualian	[8]
89	rs336977324	7	81077680	T	intron_variant	RYR3	TNB	Erhualian	[8]
90	rs80927564	7	108030803	C	intergenic_variant	-	TNB	Erhualian	[8]
91	rs80942529	7	110057727	G	intergenic_variant	-	TNB	Erhualian	[8]
92	rs80969683	7	NA				TNB	Large White	[22]
93	rs81397215	7	NA				TNB	Large White	[22]
94	rs81397142	7	NA				TNB	Large White	[22]
95	rs81403620	8	15474371	T	intron_variant	KCNIP4	TNB	Erhualian	[8]
96	rs81405013	8	16400498	C	intergenic_variant	-	TNB	Erhualian	[8]
97	rs81342198	8	16560870	C	intergenic_variant	-	TNB	Erhualian	[8]
98	rs81406385	8	17783561	C	intron_variant	-	TNB	Erhualian	[8]
99	rs81343566	8	18863902	C	intron_variant	CCDC149	TNB	Erhualian	[8]
100	rs81399474	8	32370687	C	downstream_gene_var	UCHL1	TNB	Erhualian	[8]
101	rs81399527	8	32800253	G	intergenic_variant	-	TNB	Erhualian	[8]
102	rs81399633	8	33508704	A	splice_region_variant	ATP8A1	TNB	Erhualian	[8]
103	rs81227962	8	34380373	A	intergenic_variant	-	TNB	Erhualian	[8]
104	rs81476987	8	34885027	C	intron_variant	KCTD8	TNB	Erhualian	[8]
105	rs81399897	8	36628170	C	intergenic_variant	-	TNB	Erhualian	[8]
106	rs81400131	8	38946698	A	downstream_gene_var	CWH43	TNB	Erhualian	[8]
107	rs81400868	8	63262973	A	downstream_gene_var	U6	TNB	Erhualian	[8]
108	rs81401375	8	73118551	A	intergenic_variant	-	TNB	Erhualian	[8]
109	rs339466191	8	80345448	A	intron_variant	NR3C2	TNB	Landrace/ Yorkshire	[14]
110	rs333905163	8	80589152	A	intron_variant	NR3C2	TNB	Landrace/ Yorkshire	[14]
111	rs81294311	8	88225561	T	intergenic_variant	-	TNB	Erhualian	[8]
112	rs319272490	8	103706053	C	intergenic_variant	-	TNB	Landrace/ Yorkshire	[14]
113	rs81403286	8	107008700	T	intergenic_variant	-	TNB	Erhualian	[8]
114	rs81403527	8	112530303	A	downstream_gene_var	CASP6	TNB	Erhualian	[8]
115	rs81403538	8	112566311	T	intron_variant	MCUB	TNB	Erhualian	[8]
116	rs334180816	8	115856336	C	intron_variant	NPNT	TNB	Landrace/ Yorkshire	[14]
117	rs80834695	8	120291881	A	intergenic_variant	-	TNB	Erhualian	[8]
118	rs81317149	8	127090631	G	intergenic_variant	-	TNB	Large White	[30]
119	rs81413855	9	8393770	G	3_prime_UTR_variant	C2CD3	TNB	Erhualian	[8]
120	rs81416386	9	11655571	G	intron_variant	MYO7A	TNB	Erhualian	[8]
121	rs81407589	9	21695319	C	intron_variant	-	TNB	Erhualian	[8]
122	rs81408950	9	34248222	G	intergenic_variant	-	TNB	Erhualian	[8]
123	rs81409102	9	35438101	C	upstream_gene_variant	GRIA4	TNB	Erhualian	[8]
124	rs81413928	9	84593678	A	intron_variant	AGMO	TNB	Landrace	[18]
125	rs81413949	9	85057918	C	intergenic_variant	-	TNB	Landrace	[18]
126	rs81287478	9	85128043	T	intergenic_variant	-	TNB	Landrace	[18]
127	rs81223525	9	85362483	G	intergenic_variant	-	TNB	Landrace	[18]
128	rs81260290	9	87932749	G	intron_variant	HDAC9	TNB	Erhualian	[28]
129	rs81414623	9	96283483	C	intron_variant	SEMA3A	TNB	Erhualian	[8]
130	rs81300575	9	102308556	T	intron_variant	PHTF2	TNB	Landrace	[18]
131	rs81331059	9	137095797	G	intron_variant	-	TNB	Erhualian	[8]
132	rs81419264	9	137162279	G	non_coding_transcript	-	TNB	Erhualian	[8]
133	rs81419315	9	137216947	G	upstream_gene_variant	U6	TNB	Erhualian	[8]
134	rs81315852	9	138107624	T	intergenic_variant	-	TNB	Erhualian	[8]
135	rs81332239	9	138462123	C	intergenic_variant	-	TNB	Erhualian	[8]
136	rs81417713	9	NA				TNB	Erhualian	[8]
137	rs81429231	10	10002753	G	intron_variant	MARK1	TNB	Erhualian	[8]
138	rs80895456	10	26509297	T	intergenic_variant	-	TNB	Erhualian	[8]
139	rs341908955	10	48658434	T	intergenic_variant	-	TNB	Erhualian	[8]
140	rs81329283	10	56421545	A	intron_variant	NRP1	TNB	Erhualian	[8]
141	rs81274366	10	57593745	G	intron_variant	PARD3	TNB	Erhualian	[28]
142	rs81426281	10	57913635	T	upstream_gene_variant	-	TNB	Erhualian	[8]
143	rs81314128	10	59665035	T	intergenic_variant	-	TNB	Erhualian	[8]
144	rs80941850	11	8940153	T	intron_variant	-	TNB	Erhualian	[8]
145	rs81477765	11	19542650	G	intergenic_variant	-	TNB	Erhualian	[8]
146	rs81332839	11	19542846	T	intergenic_variant	-	TNB	Erhualian	[8]
147	rs81285980	11	19544324	A	intergenic_variant	-	TNB	Erhualian	[8]
148	rs81237348	11	22281193	C	intron_variant	NUFIP1	TNB	Erhualian	[8]
149	rs80989787	11	23362846	T	intron_variant	ENOX1	TNB	Large White	[30]
150	rs81289355	11	23410214	C	intron_variant	ENOX1	TNB	Large White	[30]
151	rs81293918	11	27751988	A	intergenic_variant	-	TNB	Erhualian	[8]
152	rs80813604	11	48641897	C	intergenic_variant	-	TNB	Erhualian	[8]
153	rs80950312	11	48749391	T	intergenic_variant	-	TNB	Erhualian	[8]
154	rs80869156	11	64316157	G	intergenic_variant	-	TNB	Erhualian	[8]
155	rs81329621	11	66354904	G	intergenic_variant	-	TNB	Erhualian	[8]
156	rs81343376	12	6971812	G	intergenic_variant	-	TNB	Erhualian	[8]
157	rs81332319	12	7414272	C	intergenic_variant	-	TNB	Erhualian	[8]
158	rs81439394	12	9810809	G	intergenic_variant	-	TNB	Duroc	[29]
159	rs81433045	12	26992374	G	intron_variant	ANKRD40	TNB	Erhualian	[8]
160	rs81433877	12	33577229	G	intron_variant	MSI2	TNB	Erhualian	[8]
161	rs81434044	12	33799100	C	intron_variant	MSI2	TNB	Erhualian	[8]
162	rs81434064	12	33903675	A	intron_variant	MSI2	TNB	Erhualian	[8]
163	rs81434489	12	35993482	G	intron_variant	VMP1	TNB	Erhualian	[8]
164	rs81434499	12	36207006	G	intergenic_variant	-	TNB	Erhualian	[8]
165	rs81309004	12	40519573	T	intergenic_variant	-	TNB	Erhualian	[8]
166	rs81434931	12	40974978	A	intron_variant	ASIC2	TNB	Erhualian	[8]
167	rs81435036	12	41338150	C	intron_variant	ASIC2	TNB	Erhualian	[8]
168	rs81477883	12	43427846	T	intron_variant	RAB11FIP4	TNB	Erhualian	[8]
169	rs81447979	13	11362350	A	upstream_gene_variant	THRB	TNB	Erhualian	[8]
170	rs80837222	13	13802573	T	intron_variant	NEK10	TNB	Erhualian	[8]
171	rs81445072	13	38880472	C	intron_variant	ARHGEF3	TNB	Erhualian	[8]
172	rs80964445	13	75637294	G	intron_variant	CEP63	TNB	Erhualian	[8]
173	rs81447100	13	80866929	A	intron_variant	CLSTN2	TNB	Erhualian	[8]
174	rs81441751	13	182451198	C	intron_variant	CHODL	TNB	Large White	[30]
175	rs81442196	13	190614395	G	intergenic_variant	-	TNB	Erhualian	[8]
176	rs80961068	13	200778735	T	intron_variant	TTC3	TNB	Large White	[23]
177	rs80841768	14	12118215	T	intron_variant	PNOC	TNB	Erhualian	[8]
178	rs330621374	14	15126027	A	intergenic_variant	-	TNB	Landrace/ Yorkshire	[14]
179	rs339777110	14	25374812	G	intron_variant	TMEM132D	TNB	Duroc	[29]
180	rs334650508	14	26675179	A	intergenic_variant	-	TNB	Erhualian	[8]
181	rs318344052	14	26907303	T	intergenic_variant	-	TNB	Erhualian	[8]
182	rs80979042	14	62040147	T	intron_variant	BICC1	TNB	Duroc	[29]
183	rs80825112	14	62071531	A	intron_variant	BICC1	TNB	Duroc	[29]
184	rs80892145	14	73289391	T	intron_variant	LRRC20	TNB	Erhualian	[8]
185	rs80960182	14	84555165	A	intergenic_variant	-	TNB	Erhualian	[8]
186	rs80873788	14	85481113	A	upstream_gene_variant	RGR	TNB	Erhualian	[8]
187	rs80799654	14	98901034	T	intergenic_variant	-	TNB	Erhualian	[8]
188	rs80800806	14	104041994	T	intron_variant	IDE	TNB	Erhualian	[8]
189	rs80968475	14	107031142	T	intron_variant	SORBS1	TNB	Erhualian	[8]
190	rs80959797	14	107336340	G	intron_variant	ENTPD1	TNB	Erhualian	[8]
191	rs80938898	14	107377294	C	intron_variant	ENTPD1	TNB	Erhualian	[8]
192	rs80971725	14	107584405	C	intergenic_variant	-	TNB	Erhualian	[8]
193	rs80843720	14	107597908	T	intron_variant	ZNF518A	TNB	Erhualian	[8]
194	rs80987149	14	107725686	T	intergenic_variant	-	TNB	Erhualian	[8]
195	rs81449479	14	107794160	G	3_prime_UTR_variant	OPALIN	TNB	Erhualian	[8]
196	rs80867623	14	129792523	G	intron_variant	SEC23IP	TNB	Large White	[22]
197	rs80983641	14	137193623	G	intron_variant	PTPRE	TNB	Erhualian	[8]
198	rs80892643	14	NA				TNB	Erhualian	[8]
199	rs80782668	14	NA				TNB	Erhualian	[8]
200	rs80947288	14	NA				TNB	Duroc	[29]
201	rs80868389	15	9150979	A	intergenic_variant	-	TNB	Erhualian	[8]
202	rs81452018	15	26337909	G	intergenic_variant	-	TNB	Landace/Large White	[31]
203	rs80893476	15	82075550	C	intron_variant	MTX2	TNB	Erhualian	[8]
204	rs45435330	15	118847390	A	intron_variant	IGFBP2	TNB	Berkshire	[24]
205	rs324003968	15	NA				TNB	Commercia	[25]
206	rs81459026	16	41470716	G	intron_variant	IPO11	TNB	Erhualian	[8]
207	rs81289648	16	74903886	A	intergenic_variant	-	TNB	Erhualian	[28]
208	rs81463092	16	74971494	G	intergenic_variant	-	TNB	Erhualian	[8]
209	rs81463053	16	NA				TNB	Erhualian	[8]
210	rs331119584	17	1486143	T	intron_variant	DLC1	TNB	Erhualian	[8]
211	rs328230332	17	34329730	T	intergenic_variant	-	TNB	Duroc	[29]
212	rs81472303	18	18253338	C	intron_variant	COPG2	TNB	Large White	[22]
213	rs81471172	18	51640938	T	intron_variant	HECW1	TNB	Large White	[23]
214	rs81471381	18	53672799	G	intron_variant	SUGCT	TNB	Large White	[30]
215	rs332595701	18	53697787	G	intron_variant	SUGCT	TNB	Large White	[30]
216	rs81225238	X	15335129	T	intron_variant	PHKA2	TNB	Erhualian	[8]
217	rs80891661	X	101710962	C	intergenic_variant	-	TNB	Erhualian	[8]
218	rs322114058	X	101975674	A	intron_variant	TENM1	TNB	Erhualian	[8]

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
