# Peer review of "Survey of SNPs Associated with Total Number Born and Total Number Born Alive in Pig"

_genes, 2020, doi:10.3390/genes11050491_

Round 1
Reviewer 1 Report
The authors made a big effort to summarize information of the PigQTLdb based TNB and NBA SNPs, which should be valuable for other researchers' convinience in this filed. I have several suggestion need to be communicated.
1. NBA is the abbreviation of number born alive, but not the abbreviation of total number born alive, see line 14
2. Not sure if the keyword should be in abbreviated format SNP or GWAS, see line 27
3. Some extra space existed between words, for instance see line 48
4. Erhualan should be replaced by Erhualian on line 80
5. Seems some English words were replaced by Russia words, see line 81, 109 and so on.
6. Better use EBVs a the abbreviation of estimated breeding values.
7. It might be meaningful to add some other parameters of the identified SNP based on the literature. For instance, how much genetic variance was explained by the identified SNPs, which is very important for animal breeding industry to decide if incorporating there SNPs into selection. (Not every paper will calculate the % of genetic variance was explained by the SNPs, but I am sure some will provide).
8. It looks like even for the same trait, not many SNPs identified were shown shared across the sub-populations under a breed or across the breeds. Could the authors kindly provide some explaination for that?
9. Assuming I am a reader, I don't feel this review give me a good guidance on how to use the richful SNP information and how to find the new research direction. Would the authors considering to add one or two paragraph for it?
Reviewer 2 Report
Writing a review on SNPs associated with main reproductive performance in pigs as Total number born and Total number born alive is useful to the scientific community. Using Pig Quantitative Trait Locus Database (PigQTLdb) as source of information ensures that most of the published work on this topic has been cited. It is suggested to avoid repetitions in the text what is already reported in the tables (such as the location of SNPs on chromosomes) and to give more space to the comment on the biological effects of SNPs. I suggest rewriting the abstract to avoid it being just a repetition of some parts of the work. Moreover, the review requires a linguistic and accurate formatting revision (there are many Russian words and phrases).
In particular:
Line 39: I believe it is more correct to write reproductive performance rather than reproductive productivity.
Line 79: it is recommended to move the sentence “Ensembl db, Sus Scrofa 11.1” after the title of the table (line 78)
Line 79: it is recommended to modify “This variant has not been mapped Sscrofa 11.1” in the following way “This variant has not been mapped on Ensembl db, Sus Scrofa 11.1”
Line 85: replace “Erhualian (n = 173)” to “Sutai (n = 173)”
Line 87: replace “allele G for the Erhualian and Yorkshire” to “allele G for the Sutai and Yorkshire”
Line 89: replace “Landras” to “Landrace”
Line 94: the reference Coster et al. is the number 23 and not 22
Line 111: the reference Wang et al. is the number 22 and not 23
Line 116: it is recommended to modify the sentences: “11 Large SNPs located” in the following way: “11 SNPs located
Line 120: about reference Chen et al. (2019), replace the year with the number
Line 124: it is recommended to indicate correctly where the genes (IGFBP2 and IGFBP3) are and to rework the subsequent sentences to avoid plagiarism.
Line 135: Please change “SSC” with “Sus scrofa” because the abbreviation SSC indicates “Sus Scrofa Chromosome”
Line 156: see Line 79
Lines 164 and 167: the reference Wang et al. is the number 22 and not 23
Line 170: something is missing after "SNP rs"?
Line 211: about reference An et al. (2018), replace the year with the number
Author Response
Пожалуйста, посмотрите приложение
